# Towards Safety and Regulation Criteria for Clinical Applications of Decellularized Organ-Derived Matrices

**DOI:** 10.3390/bioengineering12020136

**Published:** 2025-01-30

**Authors:** Elena V. A. van Hengel, Luc J. W. van der Laan, Jeroen de Jonge, Monique M. A. Verstegen

**Affiliations:** Department of Surgery, Erasmus MC Transplant Institute, University Medical Center Rotterdam, 3015 GD Rotterdam, The Netherlands; e.v.a.vanhengel@erasmusmc.nl (E.V.A.v.H.); l.vanderlaan@erasmusmc.nl (L.J.W.v.d.L.); j.dejonge-1@erasmusmc.nl (J.d.J.)

**Keywords:** decellularization, extracellular matrix, tissue engineering, regenerative medicine, medical device, advanced therapy medicinal product, regulations

## Abstract

Whole-organ decellularization generates scaffolds containing native extracellular matrix (ECM) components with preserved tissue microarchitecture, providing a promising advancement in tissue engineering and regenerative medicine. Decellularization retains the ECM integrity which is important for supporting cell attachment, growth, differentiation, and biological function. Although there are consensus guidelines to standardize decellularization processes and ECM characterization, no specific criteria or standards regarding matrix sterility and biosafety have been established so far. This regulatory gap in safety, sterilization, and regulation criteria has hampered the clinical translation of decellularized scaffolds. In this review, we identify essential criteria for the safe clinical use of decellularized products from both human and animal sources. These include the decellularization efficacy, levels of chemical residue, preservation of ECM composition and physical characteristics, and criteria for the aseptic processing of decellularization to assure sterility. Furthermore, we explore key considerations for advancing decellularized scaffolds into clinical practice, focusing on regulatory frameworks and safety requirements. Addressing these challenges is crucial for minimizing risks of adverse reactions or infection transmission, thereby accelerating the adoption of tissue-engineered products. This review aims to provide a foundation for establishing robust guidelines, supporting the safe and effective integration of decellularized scaffolds into regenerative medicine applications.

## 1. Introduction

The interdisciplinary field of regenerative medicine holds great promise to improve the quality of life of patients with organ failure or damage, and by addressing organ shortage [1,2,3,4,5]. Decellularized organs are of interest to use as a scaffold for tissue engineering, as these closely mimic the native tissue environment, and facilitate cell attachment, proliferation, and differentiation [6,7]. During whole-organ decellularization, all cellular material is removed, while the unique microarchitecture, vascular network, and biochemical composition of the native extracellular matrix (ECM) is retained [7,8,9,10,11,12]. These decellularized scaffolds can be directly utilized for clinical applications as medical devices, such as porcine or bovine heart valves. Decellularized scaffolds also show potential for creating living tissue constructs. After repopulation with relevant viable cells, a functional tissue construct can be engineered that could be used for various clinical applications (Figure 1). Tissue engineering therefore has the potential to address challenges such as organ shortage by constructing transplantable organ substitutes [13,14,15,16,17,18,19]. Examples of successful tissue engineering for organ development and transplantation have been shown in preclinical models for kidney [20,21,22], liver [23], cartilage [24], trachea [25], intestine [26], and adipose tissue [27]. These approaches for tissue reconstruction show that ECM scaffolds successfully support growth, differentiation, and function of various cell types in vitro [27,28]. However, translation to clinical application has so far been hampered by the lack of clear regulations and guidelines to obtain scaffolds with consistent characteristics and quality [29]. These engineered tissues and organs could become a commercial product, as demonstrated by the existing use of decellularized porcine heart valves in heart valve replacement surgery [30]. In this case, they become medical treatments, necessitating clear and standardized regulations before this technology can be brought to the patient. Therefore, quality assurance and criteria are needed to ensure safe use, biocompatibility, and functionality of decellularized scaffolds and advance its potential in regenerative medicine [8,14,31]. Even batch variations that may influence clinical outcome should be reduced to a minimum [32,33]. Therefore, developing a clinically applicable and transplantable construct requires full understanding of all contamination hazards and includes comprehensive quality measures [13,14,31,34,35].

In this review we therefore (1) evaluate the assessment of bacterial, viral, or chemical contamination after decellularization, (2) propose detailed quality control criteria to maintain consistency and safety, and (3) assess methods to sterilize scaffolds to meet clinical-grade standards.

### Decellularization

Decellularization is a commonly used method to isolate a biological scaffold with preserved tissue or organ microarchitecture [36,37]. During decellularization, all cellular and nuclear components are removed, while the ECM and thereby its unique microstructure, properties, and biochemical cues are preserved [7,8,31,36]. Since most polymorph and immunogenic cellular components are removed, the immunological risk for alloreactivity and rejection of the decellularized scaffolds in clinical applications is minimized [14,38,39].

Depending on the type of tissue and preference of the research groups, many different decellularization protocols are being executed [40,41]. Protocols include the use of physical (e.g., freeze–thaw cycles, pressure, or sonication), chemical (detergents and solvents), and enzymatic (DNases and RNases) methods. Each decellularization technique has unique and distinct effects on the matrix, potentially altering the structural, biochemical, and mechanical characteristics [17,21,27,37,42,43,44,45,46,47,48,49]. These changes can influence the scaffolds’ biological characteristics and may affect the recipient’s immune response [2,8,9,13,33,40]. It is important that the decellularization method has a minimal effect on the ECM protein composition, architecture, and mechanical properties [42,50,51,52,53]. The most efficient decellularization approach should comprise an optimal balance between cellular removal and ECM preservation and be customized for the unique properties of the tissue or organ of interest [8,9,13,18,34,54]. Protocol standardization for decellularization of each type of tissue is essential to ensure a reliable, scalable, and reproducible production of safe scaffolds, while key ECM properties are retained, batch-to-batch variations prevented, and quality criteria fulfilled [32,33]. Decellularized products can also be processed into powders, composites, or hydrogels [38,55]. Hydrogels could be processed as bioink for bioprinting, an application that provides the opportunity to design, print and pattern a 3D matrix [56]. Currently, bioinks are mostly prepared from decellularized cartilage, adipose, and heart tissue [57,58]. The guidelines and regulations described in this review will also apply to printed constructs prepared from decellularized ECM-derived bioinks that are developed for clinical use. 

## 2. Quality and Safety Criteria

Establishing a stringent regulatory framework regarding decellularization thresholds and acceptance criteria for decellularized scaffolds is needed to ensure the production of safe ECM products and facilitate their progression towards clinical applications [8,29,33,59,60]. In 2021, the American Society for Testing and Materials (ASTM) published a consensus guide for the evaluation of decellularization processes (ASTM F3354-19) [60]. This report describes the purpose of decellularization as the removal of cells while preserving the ECM structure and composition, to generate an implantable product that will result in reduced risk on immune rejection while promoting the integration into host tissue [60]. The document does not define universal thresholds for decellularization, but solely provides guidance on the characterization and evaluation of decellularization. Acceptance criteria should, e.g., include cellular content, the retention of ECM components, and the residual of decellularization reagents and chemicals (Figure 2). Assessment of in vivo safety of medical products composed of decellularized ECM are not included, but it is recommended that testing should be performed in a relevant model, e.g., an in vivo model or ex vivo model when appropriate [35,60].

### 2.1. Characterization of Decellularized ECM

Quality criteria to describe the efficiency of decellularization should account for cellular remnants and residual nucleotides (DNA and RNA), since this highly influences the immunogenicity of the scaffold and could even result in death after implantation [14,34,61,62,63,64]. The effects of residual DNA or RNA in the scaffolds and, with that, the acceptable threshold are not well known. However, many studies describe a limit of 50 ng double-stranded DNA per mg ECM dry weight; less than 200 base pairs DNA fragment length; and no visible nuclear material after staining with Hoechst, 4′,6-diamidino-2-phenylindole, or hematoxylin and eosin [8,14,45,59,65]. This threshold is considered safe for transplantation, as described by Crapo et al. [59]. DNA fragments can negatively impact the integration of the scaffold within a recipient, and remnants of DNA in the scaffold have led to inflammation [66]. RNA nucleotides are more prone to degradation by endonucleases and exonucleases, and are often less abundant as remnant DNA [67].

Preserving the composition and architecture of the ECM is essential for successful tissue engineering, as it guides cellular behavior during repopulation [59]. Biochemical and biomechanical signals within the ECM influence cellular activities during repopulation, such as cell migration, adhesion, and differentiation. Without proper preservation, scaffolds may fail to support the desired cellular responses, affecting the achievement of functionality. Preservation of the microstructure is crucial for organ functionality: the alveolar architecture in renal ECM is essential for its structural and instructional roles, and retaining the luminal surface of blood vessels is important to facilitate endothelial cell regeneration and prevent thrombosis [68,69,70]. Different types of tissue ECM have their own unique composition and structure, with overlapping key components. The main components are structural fibrous proteins (collagens, fibronectin, laminins, and elastins) and non-fibrous proteins (glycoproteins, proteoglycans, glycosaminoglycans) [41]. The native ECM also contains growth factors, signaling molecules integrins, and metalloproteinases [38]. Since all components have their unique role in the microenvironment, tissue organization, and cell–matrix interactions, it is essential to identify the ECM composition before and after decellularization and aim to keep the ECM as complete as possible [35]. Preservation of structural proteins, such as collagens, are important for maintaining mechanical properties. Reduced levels of collagen and GAG content result in decreased tensile strength and altered structural integrity, which are crucial for tissue functionality [59,71,72]. Changes in ECM stiffness can affect the mechanotransduction of cells, and thereby affect biochemical signal and signaling pathways [73]. In a clinical setting, differences in mechanical properties between native tissue and implantable constructs could result in stress at the implant–tissue interface, or the scaffold may not withstand physiological loads, and potentially lead to mechanical failure [72]. Maintaining mechanical integrity is therefore crucial to ensure proper functionality of the decellularized product [73]. Furthermore, non-fibrous proteins and bioactive factors are vital for the functionality of decellularized scaffolds. These elements can modulate cellular behavior and promote tissue regeneration [74]. Growth factors that are embedded within the ECM can promote cellular proliferation and differentiation. If these factors are lost during the decellularization process, the effectiveness for tissue engineering and regenerative potential of the scaffold may be significantly reduced [74]. During kidney decellularization, 43% of basic fibroblast growth factor and 37% of vascular endothelial growth factor are retained, providing cues for angiogenesis and cell functionality [75]. The preservation of growth factors during decellularization is also demonstrated for liver, lung, and musculoskeletal tissues, underlining their importance in promoting tissue regeneration [23,76].

Common methodological assays for ECM characterization and identification include colorimetric assays, histological staining, SDS-PAGE, Western blot, spectroscopy, and Second Harmonic Generation [73]. No guidelines describe the acceptable levels of changes in ECM composition and levels of tissue components. The physical properties of ECM can be determined using e.g., rheology, nanoindentation, and Atomic Force Microscopy [73]. A significant decrease in ECM mechanical properties was found after decellularization of trachea, cartilage, bone, and aorta [46,77,78,79].

### 2.2. Microbial and Pathogen Contamination

To provide guidelines for safe use of medical products, sterilization methods need to be assessed on removal of microbes without damaging or altering the scaffold’s protein composition and architecture [14,43,59,80]. Since most products will in the future be developed for implantation purposes, they must meet strict sterility standards to prevent post-implant infections and immunological reactions [81,82]. Insufficient sterilization could hamper FDA approval and clinical translation [83]. In addition, mechanical properties should not be altered during sterilization, since this could influence functionality of the scaffold [31,43,80]. Traditional sterilization methods for medical devices include ethylene oxide, gamma radiation, ultraviolet radiation, and electron beam radiation [31,59,65,81,84,85,86,87]. However, it is described that irradiation sterilization methods influence the mechanical properties of the scaffold by damaging the matrix [84]. Ultraviolet radiation has very weak penetration capacities, typically a few microns, and is therefore not suitable for sterilization of larger 3D constructs [84,88]. Peracetic acid treatment is shown to be effective in sterilization, without affecting ECM proteins and mechanical properties [12,31,84]. Ethylene oxide sterilization has also been shown to effectively inactivate microbial components [84]. However, exposure to even low concentrations of ethylene oxide and peracetic acid can be harmful to people and the environment, and residue should be removed thoroughly, as described by the International Organization for Standardization (ISO) in regulation ISO 10993-7 [45]. Treatment with antibiotic/antimycotic cocktails are commonly used in tissue engineering to treat pathogen contamination. Nevertheless, its efficacy is limited, and antibiotic residue can result in antimicrobial resistance in clinical setting [59,84,89,90]. A relatively new approach is supercritical carbon dioxide, which is a promising method for sterilization of biomaterials due to good penetration capacities, without affecting the scaffold properties [14,84,91,92,93].

More research is required to set up an efficient and standardized protocol [12,31,33]. After thorough sterilization, the removal of bacteria, viruses, fungi, prions, and other pathogens needs to be confirmed to reduce the risk of pathogen transfer [94]. Ways to do this need to be specified in guidelines for clinical use of decellularized-derived products. Good manufacturing practice (GMP) standards describe that microbial, particulate, and endotoxin contamination in the final product should be below specified thresholds, and provide principles for sterilization regarding the facility, cleanroom grading, equipment, qualified personnel, monitoring systems, and packaging [95]. The sterility assurance level for terminal sterilization of medical devices is set at 10^−6^ or better, indicating a one in a million probability that a product contains a single viable microorganism [95,96]. Moreover, the inactivation of viruses and removal of viral particles and bacterial DNA are critical for ensuring biosafety. The endotoxin concentration should be assessed: endotoxin levels in medical devices should be limited to 0.5 endotoxin units per mL or 20 per device, as described by the Food and Drug Administration (FDA) guidance on sterility for medical devices and set by GMP [45,97,98]. Ensuring the sterility of the medical device is crucial to guarantee safety in clinical settings, and the appropriate sterilization method must be determined for each decellularized material, without affecting the unique ECM properties.

### 2.3. Chemical Contamination

Most protocols utilize decellularization by ionic and non-ionic detergents, such as sodium dodecyl sulfate (SDS) and Triton X-100 [59,99]. The choice of detergent, its concentration, and the administration method influences the decellularization efficiency and effect on the scaffold’s integrity. The optimal protocol should be tailored to the tissue or organ to balance effective cell removal while preserving the ECM. Triton X-100 is frequently used in decellularization due to its mild nature and effectiveness in removing cellular components, while preserving ECM structure. It disrupts lipid–lipid and lipid–protein interactions, and is effective in removal of cellular material through the formation of micelles that encapsulate and solubilize cellular debris [42]. Since Triton X-100 does not significantly affect protein–protein interactions, protocols using Triton X-100 result in better retention of the ECM’s composition and structural integrity in comparison to the ionic detergent SDS. However, Triton X-100 may be less effective for decellularization of dense tissues, and the use of stronger agents, such as SDS, is necessary for complete cell removal [100]. SDS has strong cell lysis capabilities by disrupting noncovalent bonds. However, high concentrations of SDS can denature and remove structural proteins and growth factors, and thereby disrupt the ECM’s microarchitecture. This disruption can negatively affect cell growth, adhesion, and organization during repopulation [101]. Studies have shown that when SDS is used below its critical micelle concentration (0.2%), it acts as a denaturing detergent that solubilizes the cell membrane and enables effective decellularization of small kidney biopsies, while higher levels of ECM proteins are retained [100]. 

Residual detergents may influence the biocompatibility and can cause cytotoxicity resulting in inhibiting adhesion and proliferation of cells after repopulation [84,102]. This also applies to other chemical agents often additionally used to remove cellular and nuclear debris or for sterilization of the material, including enzymes (Trypsin/EDTA, DNase, and RNase), alcohol, acids, and chemicals. Additionally, residual detergent could alter the tensile strength and composition of the ECM, and thereby result in loss of biological activity [103]. Furthermore, when implanted in vivo, residual detergent can provoke an inflammatory response and hence affect its potential for clinical applications. Extensive washing steps to get rid of detergent and chemical agents are required to minimize their levels to prevent cytotoxicity and ECM degradation [8,14,33,45]. Quantification of residual agents can be done using detection kits, spectrophotometry, or liquid chromatography [102,104,105]. Acceptable ranges of Triton X-100 are 18–70 µg/mg, and the threshold of residual SDS is 0.05 µg/mg to prevent cytotoxicity and produce a nontoxic scaffold [106,107].

### 2.4. Animal-Derived Products

Using human-derived scaffolds for large-scale applications is challenging, due to the limited availability and heterogeneity of human donor material [29,84]. Variations in donor age, health, and lifestyle result in inconsistency in the quality of available material [14,108,109]. Animal-derived scaffolds are an interesting and reliable alternative source, thanks to their widespread availability and consistent quality [31,65]. Although it is important to discuss and evaluate ethical considerations, this is outside the scope of this review.

Pig-derived scaffolds are a promising alternative to human scaffolds, due to similarities in size and physiology [44,46,110,111]. However, when using xeno-derived scaffolds, new challenges need to be addressed, including the presence of animal-derived pathogens (viruses, bacteria, or prions), species-specific antigens, and epitopes (i.e., α–gal) which could be transferred and provoke diseases or immune responses in human recipients [30,31,36,55,65,84,112]. Regarding the potential risk of viruses, this could include both exogenous animal-derived viruses which are transmissible to humans (e.g., swine flu viruses) or endogenous viruses that are encoded in the animal genome (e.g., pig endogenous retroviruses) [113,114]. Clearly, to reduce the risk of pathogen transmission, the source organism of the decellularized organs should have a specific-pathogen-free (SPF) status, implying that the source animal was free from specified pathogens (including bacteria, viruses, and parasites) for at least two years before organ retrieval [115,116]. A positive effect of decellularization could be that most animal-specific, immune-provoking components are removed [59]. Massaro et al. and Wong et al. report an overview of immunogenicity studies after implantation of decellularized xenografts in different cross-species (e.g., porcine into rat, porcine into human), presenting mixed outcomes in immune responses by the recipient [111,117]. Decellularized xenogeneic heart valves still show adverse reactions in clinical settings, including thrombosis, calcification, and rare immune responses due to the presence of α–gal antigens [118,119,120]. Heart valve retrieval from genetically modified animals could overcome the last issue [118]. To ensure safe translation of animal-based scaffolds to the clinic, thorough evaluation and monitoring is required. Extensive studies are required to reduce the risk on immunogenicity, and to achieve clinical success using animal-derived scaffolds [36,45,112].

However, multiple decellularized xenografts are already commercially available and demonstrate the safe and effective use in the clinic for porcine heart valves and artery grafts, porcine dermis, and bovine pericardium [30,59,111,121,122,123,124,125,126]. The commercially available Mosaic™ (Medtronic Inc., Dublin, Ireland) porcine-derived prosthetic implant (FDA approved in 2000) for aortic and mitral valve replacement shows positive long-term results based on rates of death, reoperation, and explant [127]. Decellularized bovine carotid artery grafts are FDA approved since 1970 (Artegraft^®^, LeMaitre Vascular, Inc., Burlington, MA, USA) and have been used in clinical settings for decades, demonstrating the potential of animal-derived decellularized scaffolds [128,129,130].

## 3. Regulatory Classification of Decellularized Scaffolds

Depending on their composition and intended use, different regulation may apply. Decellularized tissue, not containing any viable cells and functioning as a supportive framework, is considered a medical device, and therefore needs to comply to regulations as such. For example, decellularized scaffolds are utilized as structural implants, e.g., wound dressings [131], bone substitutes [132], and decellularized heart valves [33]. In the EU, medical devices are classified into four groups: Class I for low-risk and non-invasive devices; Class IIa for low- to medium-risk devices, often installed in the body for short term (maximum 30 days); Class IIb for medium- to high-risk devices, often installed in the body for longer periods; Class III for high-risk products, for implantation and/or to support life (e.g., pacemakers, breast implants) as summarized in Figure 3 [133,134,135]. In the United States of America (USA), Class IIa and IIb are merged and considered as Class II [133].

The international standard for medical devices is drafted by the ISO and describes the requirements for designing, developing, producing, installing, and servicing medical device (ISO 13485:2016) [135]. These standards are subjective to additional national and continental specific regulations and frameworks: For implantable and Class III medical devices on the European Union (EU) market, the regulatory requirements described by the EU Medical Device Regulation (MDR) should be met (MDR, 2017/745 and 2024/1860) [136,137,138,139]. The CE (Conformité Européenne) mark is affixed to a medical device when it complies with all relevant European regulations and has passed the conformity assessment, confirming its legal placement on the EU market [140]. In the USA, medical device regulations are coordinated by the FDA Center for Devices and Radiological Health [135].

When decellularized scaffolds are repopulated with viable cells, the construct is classified as an Advanced Therapy Medicinal Product (ATMP). Products that are included as ATMPs contain engineered cells or tissues; consist of non-viable human or animal tissues with a pharmacological, immunological, or metabolic intended function; or are a combination of non-viable tissue with a medical device [82,141]. The European Medicines Agency (EMA) and its Committee for Advanced Therapies (CAT) is accountable for the scientific evaluation and classification of new ATMPs, and the monitoring of safety, quality, and efficacy after approval [14,142]. The specific classification depends on whether the construct is used for structural support, or actively contributes to tissue regeneration [143]. Compared to traditional medical products (e.g., catheters, artificial pacemakers, and breast implants), ATMPs require distinct regulations and procedures for clinical translation [144].

Next to gene therapy medicines and somatic cell therapy medicines, tissue-engineered products (TEPs) are a subtype of ATMPs [143,145]. TEPs are described as a construct, containing cells or tissues, with the aim to be integrated in the human body to repair, supplement, reconstruct, or replace dysfunctional tissues or organs [144,146]. By mimicking natural tissue architecture and functionality, TEPs are worldwide investigated to address complex medical challenges, to improve the outcomes for patients that require complex tissue replacements and to solve organ shortage [4]. Despite their clinical potential, most research is still in the early stages and only a limited number of products have reached (pre-)clinical trials and transition to the market. Successful examples include decellularized bladder matrix that is used in a clinical setting for patients needing cystoplasty [147], decellularized small intestinal submucosa sheets for cardiac repair and reconstruction [148], and decellularized pulmonary grafts [149,150]. In 2014, the first ATMP construct containing stem cells was recommended for approval by the EMA. This was a bioengineered corneal epithelium containing autologous stem cells to repair damaged corneal surface (Holoclar®, Holostem, Modena, Italia) [144,151]. The difference between the regulatory frameworks in the European Union and the USA for medical devices and TEPs, as subcategory of ATMPs, is summarized in Figure 4.

Based on the acceptations and predictions the EMA and the global Alliance for Regenerative Medicine, it is expected that the development of ATMPs will further accelerate over the coming decade [145,152,153]. Tissue-engineered products that have recently been classified by the EMA [143] include:

Skeletal muscle-derived cells attached to microparticles for treatment of incontinence and anorectal malformation;Adipose-derived stem cells seeded into a polypropylene conduit mimicking the extracellular environment of the urinary tract, for radical cystectomy in bladder cancer;Decellularized dermal scaffold repopulated with allogeneic human Wharton’s jelly-derived mesenchymal stem cells, for the treatment of epidermolysis bullosa.

For medical devices designed and developed in the USA, the FDA describes the regulations for the use of medical products [97]. Multiple TEPs have been FDA-approved for use in clinical settings, demonstrating the potential of acellular matrix products in medical healthcare. Examples that have been regulatory approved include human dermis (Alloderm^®^, LifeCell, Corp., Branchburg, NJ, USA); small intestinal submucosa grafts (SurgiSIS^®^, Cook Biotech, Inc., West Lafayette, IN, USA; Restore^®^, DePuy Orthopaedics, Inc., Raynham, MA, USA); urinary bladder matrix (ACell Inc., Columbia, MD, USA) and heart valves (Synergraft^®^, Artivion, Kennesaw, GA, USA); and epicardial extracellular matrix patches for cardiac repair (2014) and tricuspid valve replacement (2019) (CorMatrix^®^, Inc., Roswell, GA, USA) [8,38,84,148,154]. Products containing human cells, tissues, and cellular- and tissue-based products (HCT/Ps), used for implantation or infusion, are regulated via the Regenerative Medicine Advanced Therapy [155].

Asia does not have a centralized regulatory body, and each country has its own association responsible for establishing regulations: the National Medical Product Administration for China [156]; Japan’s Pharmaceuticals and Medical Devices Agency [157]; the National Medical Device Policy for India [158]; and the ASEAN Medical Device Committee for Southeast Asian Nations including Singapore, Thailand, Indonesia, and the Philippines [159]. The Australian Regulatory Guidelines for Medical Devices and Therapeutic Goods Administration (TGA) are responsible for the regulation and classification, respectively, of medical devices and advanced therapies in Australia [160,161]. Medical and tissue-engineered products are classified and regulated similar to drugs by Health Canada [162].

## 4. Regulatory Framework

The good manufacturing practice (GMP) is a collaborative effort between international and national regulatory organizations, initiatives, and industry experts, including the World Health Organization (WHO), FDA, and EMA. GMP standards are set for food, cosmetics, pharmaceutical products, dietary supplements, and medical devices, and provide the minimum requirements that a manufacturer must meet to assure that the product is consistent from batch-to-batch high in quality, are appropriate for their intended use, and meet the requirements for marketing or clinical trial authorization [163]. The principles and guidelines of GMP in the EU cover, e.g., the quality system and control, documentation, and sterility [95,163]. GMP compliance is required for the production and distribution of clinical-grade products [164].

Currently, regulations are available for using medical devices in clinical applications, with specific guidelines for products derived from biological materials. In Europe, medical devices are regulated by the EU Medical Device Regulation (MDR 2017/745), specifying safety, efficacy, and mechanical support of the product [140]. The EU Medical Device Regulation (MDR, 2017/745 and 2004/23/EG) applies to medical devices made from non-viable tissue, including decellularized materials, but does not apply to constructs containing viable cells [140]. ISO guidelines are described to meet quality and efficiency goals and are therefore relevant during the manufacturing of medical devices and products (Table 1). The ISO/TC 150/SC 7 is a subcommittee of the International Organization of Standardization (ISO) and focuses on regulations for biological evaluation of medical devices and TEPs. The most important standard for TEPs, as described by ISO/TC 150/SC 7, is the ISO 10993 framework that includes biocompatibility tests, biological evaluation, risk analysis, cytotoxicity assays, and chemical analysis [45]. The primary aim of ISO 10993 is the protection of humans from potential biological risks arising from the use of medical devices [165]. Other relevant subcommittees of the ISO that drive standardization of medical devices include ISO/TC 194 (Biological and clinical evaluation of medical devices) [166], ISO/TC 210 (Quality management for medical devices) [167], and ISO/TC 212 (Clinical laboratory testing and in vitro diagnostic testing) [168]. For a product containing tissues or viable cells that are incorporated in a medical device, the regulations are determined by the mode of action of the construct. Accordingly, the immunogenicity, pharmacological and metabolic activity of the cells are crucial for complying with the ATMP-specific regulations and to be reproducible for clinical grade [31,144].

The development and approval of ATMPs within the EU is described by Regulatory Framework EC 1394/2007, providing the regulatory considerations for tissue-engineered products and combined ATMPs, including Directive 2001/83/EC describing the safety, efficacy, and quality of medicinal products for human use; EC 726/2004 for authorization and supervision of medicines for human use; and Directive 2009/120/EC representing the scientific and technical requirements [144,164,169,170]. Directive 2004/23/EC needs to be considered when tissues and cells are used in a medicinal product, describing the standards for donation, procurement, testing, processing, preservation, storage, and distribution of the product. Furthermore, technical requirements, traceability, coding, and standards for quality and safety are described [14,144]. In the USA, the regulations for human cells, tissues, and cellular- and tissue-based products (HCT/P(s)) are presented in 21 CFR 1271, reporting the assessment on screening, donor testing, packaging, processing, storing, labeling, and distribution of the products, which also incorporates the applicable ISO standards [171]. Furthermore, 21 CFR 820 describes the requirements for quality system regulation of medical devices [172].

## 5. Preclinical and Clinical Testing

Preclinical and clinical testing, including phase I, II, and III trials, are essential to ensure the safe and effective use of decellularized products and TEPs for clinical application [73,173]. Extensive preclinical studies, including in vitro and in vivo assessments in relevant animal models, are required for evaluation of toxicity, microbial safety, immunogenicity, functional performance, and biocompatibility [14,65,174]. For all parameters, foremost for testing immunogenicity, relevant models must be used and risk management should be performed during all stages of the study [31,175]. It is thought that the main contributing factors for the host immune response against ECM scaffold are decellularization efficiency, ECM alteration, and the site of implantation. The immunogenicity of ECM scaffolds is extensively described by Kasravi et al. [39], as well as the challenges that need to be overcome and that a standard for acceptable immunogenicity should be established.

In vitro testing provides valuable information about the ECM scaffold’s biocompatibility, its degradation profile, and effect on the viability, metabolic activity, and differentiation of cells [73]. Although in vitro models have a limited translation to living organisms and limited clinical relevance, these models are essential to pave the road towards in vivo models, aimed at evaluation of safety and efficacy. Newer techniques, such as ex situ machine perfusion may provide innovative platforms to test many important aspects of a construct, respecting animal welfare, and reducing associated experimental costs and regulatory issues [176].

Nowadays, many products are being studied in in vivo, including research on bioengineered vessels for vasculature repair or replacement. However, due to the complexity of the vasculature network and vascularization, many vessel grafts remain at the preclinical stage [177]. Promising data is however shown by Humacyte, Inc., Durham, NC, USA, which has developed small-diameter human vascular grafts, demonstrating its potential in preclinical applications for treating patients with advanced peripheral artery disease [178,179]. The company received the FDA Regenerative Medicine Advanced Therapy designation [179]. Furthermore, human acellular vessel grafts for patients with renal disease are currently under investigation in phase I/II trials. First results show functional hemodialysis without infection after implantation [110,178]. A phase I/II study using porcine-derived myocardium ECM hydrogel for myocardial injection showed positive effects, proving the safety and feasibility of ‘VentriGel’. Improved remodeling of the left ventricular one year post-injection in 15 patients was demonstrated [180]. Furthermore, LifeNet Health, Virgina Beach, VA, USA, is currently recruiting patients for a phase III trial to investigate the clinical efficacy of a human placental membrane as treatment for diabetic foot ulcers. The placental membrane is decellularized and terminally sterilized using gamma irradiation in pursuit of acceptation for surgical applications [181,182].

Clinical trials are essential to validate safety and efficacy, to test immune response upon implantation and assess the occurrence of thrombosis [29,64,183]. An important lesson was learned from the use of SynerGraft (Artivion™) decellularized human heart valves. These were FDA approved in 2008 for congenital cardiac surgery and aortic valve replacement. However, in 2001, these decellularized heart valves resulted in the death of three European patients due to early complications, caused by incomplete removal of cellular components [61,62]. Pre-clinical tests were performed using sheep models, but clearly failed to predict the immune response in humans. This caused a stop to implant the SynerGrafts valves [62]. In 2004, the FDA application for the SynerGraft valve was terminated after microorganisms were detected in the decellularized tissues [83]. The cleaning process was subsequently improved with antibiotic treatment and sequential washing [184]. Recent studies show no deaths due to structural failure of the SynerGraft valve in a patient cohort of 163 patients at an intermediate- to long-term follow-up after implantation [150]. The case of SynerGraft shows that during preclinical testing not all potential complications can be anticipated and highlights the importance of strict monitoring and extensive testing before clinical approval.

Extended phase I trials are required to thoroughly assess the safety profile of the scaffold and observe adverse events. Careful and comprehensive evaluation in phase I is critical to minimize safety issues emerging during phase III trials. Subsequently, phase III randomized controlled trials will provide robust evidence regarding the efficacy and performance of the medical device [185]. The standard for medical devices for human subjects (ISO 14155:2020) describes requirements to protect human subjects; ensure the scientific conduct of the clinical investigation studies; define responsibilities of those involved; and assist those involved [186]. Post-market surveillance, including clinical follow-up activities, ensures that the device continues to meet safety and performance standards on long-term [98,122,175]. After completing preclinical and clinical testing, an application for FDA approval can be submitted, including all data from preclinical test to phase III trials, demonstrating the safety and effectivity of the product [173]. In the EU, clinical evaluations must include comparisons with the state of the art and alternative treatment options, using published literature and clinical trial data [139]. The MDR has stricter requirements compared to the US, making the approval process for Class III devices prolonged and more challenging. However, clinical data gathered in the US can accelerate the EU approval process [139].

### Clinical Successes

Human- and animal-derived decellularized materials are widely evaluated for clinical applications, and multiple products are already commercially available, demonstrating their potential in clinical use [59,111,121,122,123,124,125,126]. In particular, dermal and bone grafts, which are relatively easily sterilized, are FDA-approved at large scale [31]. Decellularized bone-derived products for orthopedic surgery purposes (e.g., GraftCage^®^, Johnson&Johnson, NJ, USA, Puros^®^ DBM, ZimVie, Florida, USA, InterGro^®^, Clearwater, FL, USA) have been successfully and safely implanted in large numbers since 1985 [31,187,188,189]. Similarly, FDA approved dermal grafts have shown to promote skin healing and wound healing by supporting keratinocyte and fibroblast repopulation (Oasis wound matrix, Alloderm^®^, and GraftJacket^®^) [31,61,131]. Examples of clinically available products for cardiovascular purposes include both human and animal-derived heart valves (CardioCel^®^ by Admedus IHS Inc., London, UK, Medtronic Inc., and Perimount by Edwards Lifesciences Corporation, Irvine, CA, USA), carotid arteries for bypass (Artegraft^®^, LeMaitre, Inc.), and epicardial patches for cardiac surgery (CorPatch^®^ by CorMatrix^®^) [31,61,84,111,190]. Host cell repopulations of acellular valves have shown successful integration with recipients [31].

## 6. Discussion and Conclusions

Innovations in decellularization technologies provides biological scaffolds for tissue regeneration and clinical applications [31,36]. Clinical use requires efficient and robust decellularization procedures and standardized guidelines [33,35,37,38]. Decellularization guidelines should incorporate the preservation of biological and mechanical properties of the ECM and guide sensitive testing of remaining cellular components and characterization of ECM components, e.g., quantification of major ECM components, including the total collagen and glycosaminoglycan levels [60,174]. Furthermore, mechanical evaluation and chemical contamination should be assessed. Sterilization of ECM is required to remove the microbial load and meet the clinical accepted sterility assurance level of 10^−6^ and accepted endotoxin levels [38]. However, guidelines for standardization of decellularization and sterilization should be organ- or tissue-specific, considering the retention of the unique characteristics and composition of the ECM.

The repopulation of the scaffolds with cells is a critical process and was not included in this review. Repopulation requires a different regulatory framework that includes detailed information on suitable cell sources, complete donor screening, cell characterization, and viability and functionality testing to ensure reproducibility and safety [14,31,41,174,191]. With standardization of procedures, more efficient, reliable, and reproducible experimental studies can be performed, resulting in faster clinical employment of these devices [33,82]. Current clinical successes are limited to small-sized products derived from decellularized tissues. The transitioning to full-sized organs will require a stricter and clearer regulatory framework to guarantee safety, efficacy, and functionality post-implantation. Standardized protocols for manufacturing, matrix characterization, and quality control are crucial to accelerate clinical translation and will allow faster transfer availability to the patient [33,38,174]. All steps from scaffold development to implantation should be closely monitored: logistics, preservation, transport, identification of intermediate products, and understanding the immunological reaction by the host [31]. Biocompatibility and immunogenicity evaluations are essential for clinical implementation [174]. Robust testing in relevant ex vivo or in vivo model is required to address the immune response and ensure safety of the product [4]. Guidelines for preclinical and clinical trials are required and should align with the set ISO standards for medical devices. The gold standard for controlled, randomized, and double-blind clinical trials may not be feasible or ethically approved in all settings, but a strict regulatory framework should be set [169]. Phase I/II trials for extensive safety evaluation should be prioritized to prevent disseminating decellularized technologies into uncharted territory, as happened with SynerGraft decellularized heart valves in the past.

Transitioning from laboratory-scale production to clinical grade-products remains a significant hurdle to take [14,33,192]. Compliance with FDA end EMA standards is important to meet both regulatory bodies and facilitate translation in the USA and European countries. The regulatory landscape is complex, often country-specific, and the standard pathway for the regulatory approval approximately takes 10 months [40,161]. Nevertheless, even after certification and acceptation by the FDA and ISO, several adverse reactions to the clinically used implants are still reported [33,45]. To ensure that products are safe and effective for patients, GMP and extensive clinical trials with long-term follow-up and risk management studies are required to engineering a clinical-grade product [31,169]. Especially for ATMPs, procedures and methods should be regulated throughout the whole process: from collecting the starting material, decellularization standards to ensure safety and quality, and product storage, to supplying the finished product within its shelf-life to the recipient [169].

The success of translation of decellularized products to the clinic relies on overcoming scientific, logistic, and regulatory challenges. Increased standardization will not only enhance reproducibility, but also facilitate faster translation and result in unlocking the full potential of tissue engineering. With consistent advancements, decellularized-derived products can substantially improve patient outcomes in the years to come.

## Figures and Tables

**Figure 1 bioengineering-12-00136-f001:**
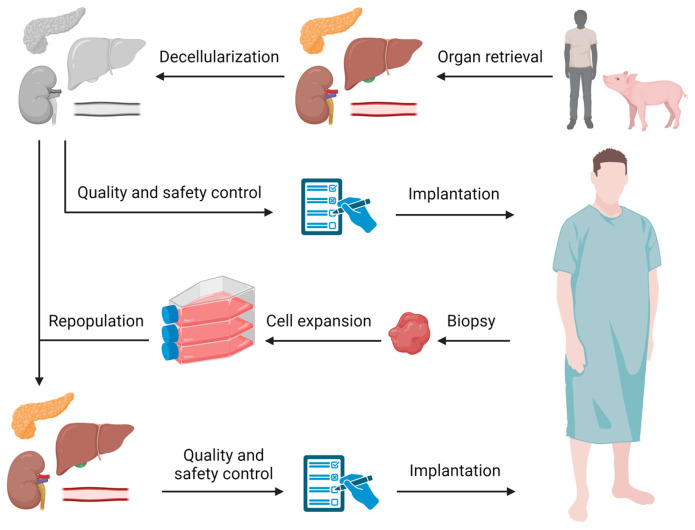
Regulatory paradigm for tissue engineering. Human or animal-derived organs can be collected and decellularized for regenerative medicine applications. These acellular products can be directly implanted into the patient (in vivo) to act as supportive material, or the decellularized scaffold may be the basis of an ex vivo tissue-engineered construct. Directly implanted acellular products such as heart valves and decellularized bone substitutes are considered a medical device and need to be regulated as such. Decellularized scaffolds repopulated with viable cells are classified as Advanced Therapy Medicinal Product (ATMP). For this, cells of the potential recipient will be isolated and expanded to be used to repopulate the decellularized scaffold according to good manufacturing practice (GMP). For both applications, a regulatory framework and quality assessment is required before implantation.

**Figure 2 bioengineering-12-00136-f002:**
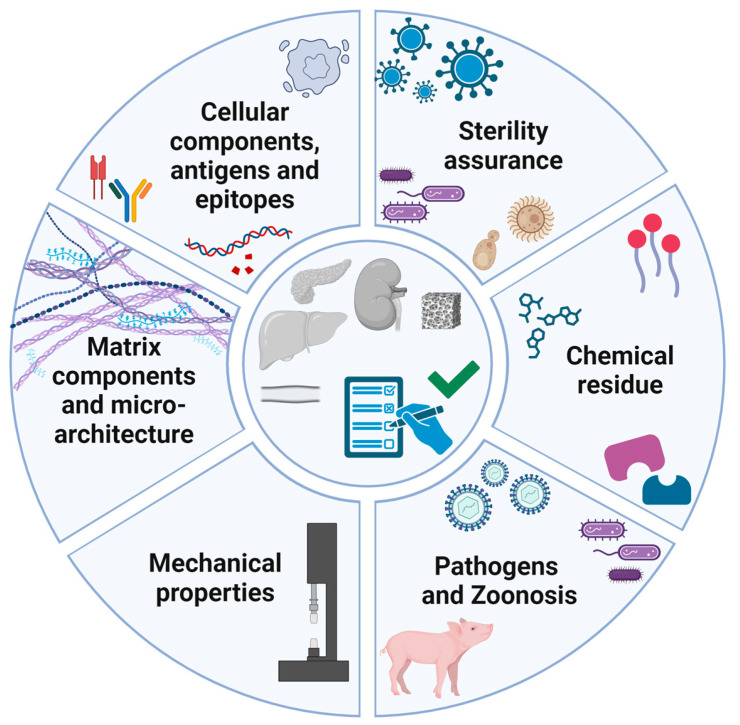
Quality and safety criteria for translation of decellularized scaffolds to clinical applications should include the presence of cellular residue and DNA fragments; the content of matrix components; the effect of decellularization on the mechanical properties; the presence of microorganisms; the residue of decellularization detergents and other chemicals; and, when using animal-derived tissue or organs, the presence of pathogens (viruses and bacteria) and animal-specific antigens or epitopes. Zoonosis are infectious diseases that are transmitted from a non-human animal to humans.

**Figure 3 bioengineering-12-00136-f003:**
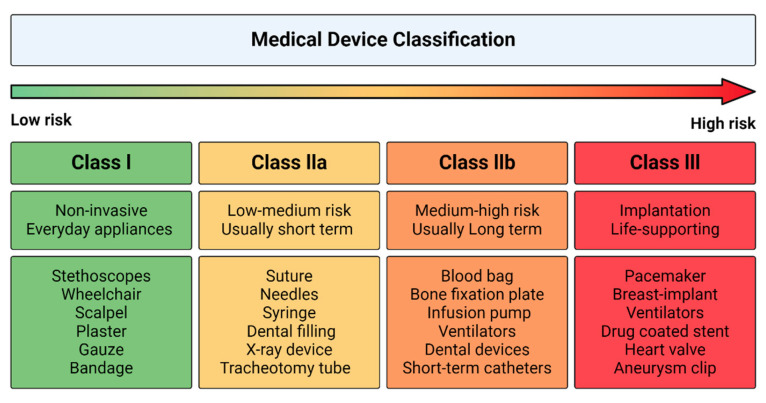
Medical device classification, based on the risk of illness or injury (low to high risk) that could be initiated by the medical device. Examples of medical devices per class are listed.

**Figure 4 bioengineering-12-00136-f004:**
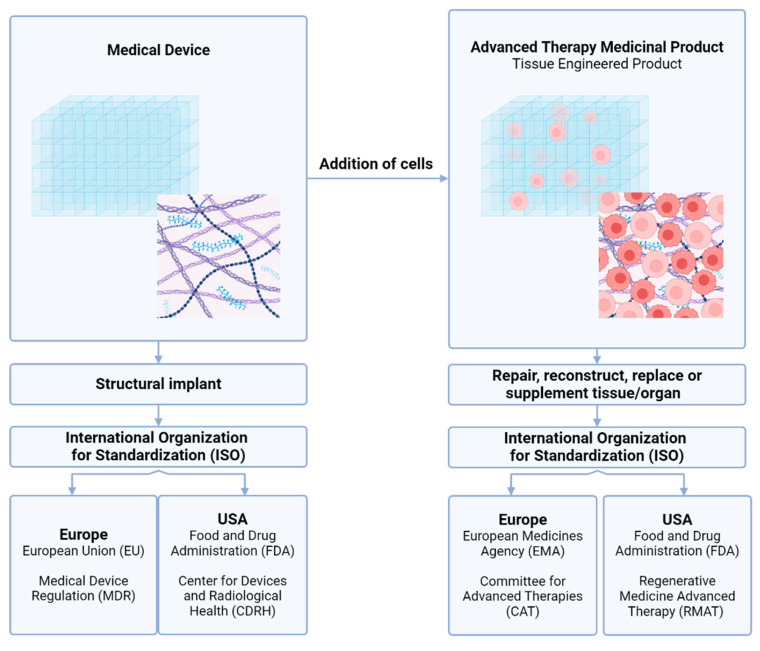
A supportive framework, utilized as a structural implant, is classified as a medical device (Class III) and regulated by the Europe Union by the EU Medical Device Regulation (MDR). In the USA, this is coordinated by the FDA and Center for Devices and Radiological Health. When cells are grown in, on, or encapsulated by a carrier material, with the aim repair or replace tissue, the product is classified as a TEP, a subcategory of ATMPs. In Europe, TEPs are regulated by the EMA and assessed by the Committee for Advanced Therapies (CAT). For the USA, assessment is performed by the Regenerative Medicine Advanced Therapy as part of the FDA.

**Table 1 bioengineering-12-00136-t001:** Framework and standards as described by the International Organization of Standardization, relevant for decellularized-derived medical products.

Standard	Content
ISO 10993	Biological evaluation of medical devices.
ISO 11607	Provides guidance on ensuring sterility of the device during the distribution cycle.
ISO 13022	Risk acceptability, expected medical benefit, available alternatives, and requirements and guidance on risk management for medical devices, medicinal products, and active implantable devices, or combinations of these.
ISO 13485	Medical devices—Quality management systems—Requirements for regulatory purposes.
ISO 14155	Clinical investigation of medical devices for human subjects—Good clinical practice.
ISO 14630	Specifies the general requirements for performance, design, materials, design evaluation, manufacturing, sterilization, packaging, and testing to prove compliance with these requirements.
ISO 14971	Describes risk management during design and development process of medical device.
ISO 18362	Manufacture of cell-based health care products—Control of microbial risks during processing.
ISO 21560	General requirements (safety, quality control, biological effects) for TEPs in regenerative medicine. Clinical trials and efficacy are not included.
ISO 22442	Medical devices utilized with animal-derived materials. Describes how to identify the associated hazards and specifies the risk management process and contamination risks.

## Data Availability

Not applicable.

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
