# Peer review of "Towards Safety and Regulation Criteria for Clinical Applications of Decellularized Organ-Derived Matrices"

_bioengineering, 2025, doi:10.3390/bioengineering12020136_

Round 1
Reviewer 1 Report
Comments and Suggestions for Authors
This manuscript aims to present regulatory guidelines for the manufacturing of acellular ECM products for clinical use. They mention that the use of aECM is still in the experimental stage although some products such as heart valves, and bladder have been in the market for many years now. The field of developing aECM products is moving towards more clinical grade products coming on line and having a set of guidelines for their production to ensure a safe product is important. One issue I have is that guidelines based on current protocols and techniques can hinder the development of improved new products as guidelines or regulations are difficult to modify once put into place. So I appreciate that in this manuscript they talk about how decellularization of organs and tissue requires very different protocols depending on the characteristics of the tissue or organ being decellularized and this must be taken into consideration.
The paper is well written and easy to follow. My main criticism is that it does not provide strong arguments for the various guidelines that are recommended. Also, there is a good section on current regulatory bodies but no information on whether they are doing a good job already or if they are not and new regulations are required. The manuscript lacks an argument for or against the current regulations already in place for decellularized ECM products.
Throughout the paper the authors mention aspects of decellularization that should be considered for regulation such as residual detergent, DNA, chemical and cell material. For each of these parameters the authors provide wide acceptance parameters. This makes the paper very passive and they do not really argue for or against any specific guidelines. The last part of the review talks about existing regional regulatory bodies and their regulations already in place. These regulatory bodies already have guidelines for ECM products used clinically so the recommendations of this review are somewhat redundant. The authors of this manuscript do not comment if the current standards are acceptable or not. Are they too stringent or too loose or too vague? Is there any evidence that the current World Health Organization (WHO), FDA, and EMA standards are not working? Are products inferior and are patients at risk? Do we even need a separate set of regulations? The paragraph on the Synergraft clinical trial supports that current regulatory agencies do not always get it right, but they do not offer any alternative. The Synergraft paragraph ends with the fact the current regulatory body made recommendations that were implemented and lead to improvements in the product.
In the discussion the authors reiterate there opening statement that standards are required for acellular ECM products but in between they really do not offer any solid advice beyond describing all of the processes in a decellularization protocol and vague suggests that each process could have guidelines. The current system of regulatory bodies seems to be working and nothing solid is offered in this manuscript. The current manuscript provides a great framework for presenting the multiple areas that should be considered for regulations or standards. I think more focus on what those regulations should be and a stronger statement followed by a justification of a specific set of ranges for residual elements would improve this manuscript. Even a strong argument that the current regulatory bodies are already meeting the needs with examples would also be great.
Minor: The manuscript mentions concentrations and types of detergents used in decellularization as being important considerations. An important point they miss is the underlying reason why different detergents and concentrations are considered. More detail on the chemical properties of detergent interactions with proteins such as the critical micelle concentration would benefit the manuscript. Two reference examples are:
1. Bongolan T, et al. Decellularization of porcine kidney with submicellar concentrations of SDS results in the retention of ECM proteins required for the adhesion and maintenance of human adult renal epithelial cells. Biomaterial Sci. 2022 May 31;10(11):2972-2990. doi: 10.1039/d1bm01017d. PMID: 35521809.
2. Dettin et al, Biomed Res Int. 2017 Jun 6;2017:9274135. doi: 10.1155/2017/9274135
Natural Scaffolds for Regenerative Medicine: Direct Determination of Detergents Entrapped in Decellularized Heart Valves. PMCID: PMC5476881 PMID: 28676861
Reviewer 2 Report
Comments and Suggestions for Authors
Comments:
1. The authors should also include the use of dECM for bioprinting applications in the Introduction with relevant references.
2. What is the source of these organs? Human-based or animal-based? The authors need to provide more in-depth explanation on this and compare the advantages and limitations of these 2.
3. What are some of the important considerations for preserving the intricate features within the organs?
4. Does the dimensions of the decellularized organs matter?
5. What are some of the reported clinical studies using decellularized organs? It would be good if the authors can include some case studies.
Author Response
Please see the attachment 'Cover letter for the Reviewers'

Round 2
Reviewer 1 Report
Comments and Suggestions for Authors
The revision covered my suggestions.
Reviewer 2 Report
Comments and Suggestions for Authors
The authors have addressed all the comments, the revised manuscript can be accepted in present form.